# Association of self-reported snoring and hyperuricaemia: a large cross-sectional study in Chongqing, China

Ting Chen ,[1,2] Xianbin Ding,[2] Wenge Tang,[2] Liling Chen,[2] Deqiang Mao,[2] Lingling Song,[1] Xuemei Lian[1]

[1]School of Public Health and Management, Chongqing Medical University, Chongqing, China
[2]Departement of Non-communicable Disease Control and Prevention, Chongqing Center for Disease Control and Prevention, Chongqing, China

**Correspondence to**
Professor Xuemei Lian;
xuemeilian@cqmu.edu.cn

## ABSTRACT

**Objective** To examine the relationship between self-reported snoring and hyperuricaemia in a large-scale population in Chongqing, China.

**Setting** Face-to-face electronic questionnaire survey, physical examination and biological sample testing were conducted in 13 districts of Chongqing. Chongqing is a municipality in southwest China.

**Participants** In this study, 23 308 Han ethnicity permanent residents aged 30–79 years were recruited. Individuals missing data were excluded, 22 389 subjects were included in final analysis.

**Primary and secondary outcome measures** Serum uric acid (UA) was measured using an oxidase method. Hyperuricaemia was defined as serum UA >420 µmol/L in men and >360 µmol/L in women. Information about self-reported snoring was obtained by questionnaire survey. All participants were divided into 'no snoring' 'snoring occasionally' and 'snoring frequently'. Multivariable logistic regression analysis was performed to assess the relationship between self-reported snoring and hyperuricaemia.

**Results** The prevalence of hyperuricaemia was 14.43%, and snorers were more likely to have hyperuricaemia than non-snorer in different age and gender groups. For the total population, those who snore occasionally or frequently were more likely to be hyperuricaemia (OR 1.19, 95% CI 1.07 to 1.31; OR 1.33, 95% CI 1.19 to 1.47) compared with no snoring people. Stratification by age, gender and body mass index (BMI), we found that the positive association between snoring frequently and hyperuricaemia was insisted in different age, gender and high BMI groups, and the strength of association varied with different age, gender and BMI category.

**Conclusion** Snoring frequency was positively associated with higher risk of hyperuricaemia. Snoring frequently may be a signal for hyperuricaemia, especially for women, those over 59 years of age, or those who are overweight or obese.

## INTRODUCTION

Hyperuricaemia is a common chronic disease caused by purines metabolism dysfunction and uric acid (UA) excretion disorder. Hyperuricaemia is becoming an important public health issue with its increasing prevalence in China. A meta-analysis study including 177 eligible studies between 2000 and 2019 reported that the prevalence of hyperuricaemia was 16.4% in mainland China.[1] Previous studies have confirmed that hyperuricaemia not only is associated with the development of gout but also increases risks of vascular diseases, diabetes mellitus, kidney disease and metabolic syndrome.[2–7] Given the large disease burden related to hyperuricaemia,[8] it is necessary to recognise more risk factors correlated with it.

Meanwhile, various major risk factors for hyperuricaemia have been found, such as low economic status,[9] obesity[10] and smoking,[11] and some studies have found a link between sleep-disordered breathing and hyperuricaemia.[12–17] However, the association of snoring and hyperuricaemia has been studied much less extensively. To our knowledge, only three studies[12 15 16] have been referred to explore the relationship of snoring and hyperuricaemia. In addition, the three studies had small sample sizes, considered confounding factors insufficiently and did not examine the association by age or body mass index (BMI) category stratified analysis.

This study aimed to examine comprehensively the relationship of self-reported snoring and hyperuricaemia after controlling for potential confounding factors in a large-scale population in Chongqing, China.

## METHODS
### Participants
The present study was from the baseline survey of the China Multi-Ethnic Cohort Study,[18]

---

**Strengths and limitations of this study**

► The sample size of the study was relatively large.
► This study analysis was thorough and comprehensive.
► The cross-sectional data analysis limited the ability to explore the casual relationship.
► Our study results were limited by age and ethnicity.
► Snoring frequency data were subjective.

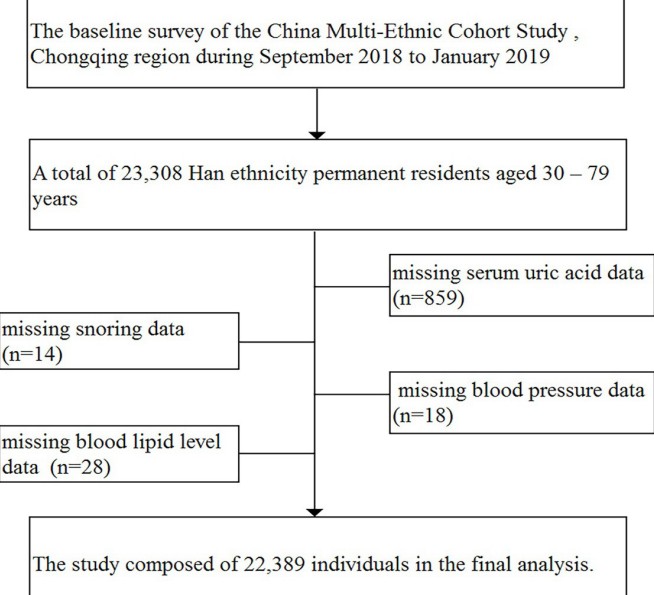

**Figure 1** Data cleaning flowchart.

Chongqing region, which is a collaborated study by Sichuan University, Chongqing Medical University and Chongqing Center for Disease Control and Prevention. This study was carried out in 13 districts of Chongqing, including Yuzhong District, Jiulongpo District, Nan'an District, Ba'nan District, Changshou District, Jiangjin District, Hechuan District, Qijiang District, Dazu District, Tongnan District, Rongchang District, Wulong District, Fengdu District. In brief, a face-to-face electronic questionnaire survey, physical examination and biological sample testing were conducted in this study. In order to ensure the quality of research, strict quality control measures were applied, for example, checking electronic recording. A total of 23 308 Han ethnicity permanent residents aged 30–79 years were recruited by using randomised sampling from September 2018 to January 2019. The sample inclusion criteria and exclusion criteria have been described in detail previously.[18] We excluded individual data missing serum UA (n=859) or snoring (n=14) or blood pressure (n=18) or blood lipid level (n=28). Finally, this present study consisted of 22 389 (figure 1) individuals providing complete information. All participants had written informed consent prior to the study.

## Measures
### Serum UA measurement and the definition of hyperuricaemia
After fasting for 12 hours, venous blood samples were collected from all participants. Within 2 hours after collection, blood samples were transported in dry ice to one medical laboratory certified by Chongqing Center for Disease Control and Prevention. Serum UA was measured with a Beckman Coulter AU 680 device using an oxidase method in Chongqing Di,an Medical Laboratory Center. The validity of serum UA assessment was described by relative deviation, which was controlled within 10%. The coefficient of variations of serum UA assessment was ≤4.0% within the group.

Hyperuricaemia was defined as serum UA >420 µmol/L in men and >360 µmol/L in women.[9 19]

### Measurement of snoring
Information about self-reported snoring was obtained by the question 'Do you snore when you have a sleep?' Then, the frequency of snoring ('occasionally' and 'frequently') was asked for those who answered 'yes'. Finally, all participants were divided into 'no snoring' 'snoring occasionally' and 'snoring frequently'.

### Assessment of covariates
The present study included sociodemographic characteristics such as age, gender (male, female), marital status, education level, residential region (urban, rural), employment status (yes, no), household yearly income level (<60 000 RMB, ≥60 000 RMB). Consistent with previous studies in China,[20 21] participants were grouped into low-aged adults (<45 years), middle-aged adults (45–59 years) and older adults (>59 years). Marital status was categorised into married or living with partner and others, including widowed, unmarried, divorced or separated. Education level was classified into primary or below (elementary school or illiterate), secondary (junior school, senior school or secondary vocational school) and college or above.

Moreover, the present study collected information about health-related behaviours. Smoking status was grouped into never smoker, current smoker and ever smoker. Alcohol drinking status was categorised into never or hardly drinker, occasionally drinker (only drink on special occasions or the frequency less than once a week), often drinker (at least once a week in the past year). Tea and sugar-sweetened beverage drinking status were defined by the question 'Have you ever been drinking tea/sugar-sweetened beverage weekly for more than half a year?' Accordingly, participants were divided into drinker and non-drinker. Oil and salt intake were collected by asking about the consumption of oil and salt per month in the past year. Energy intake was measured using a semi-quantitative standardised food frequency questionnaire with 13 food categories, including red meat, aquatic/sea food products and so on. Red meat intake was transformed into four categories (≤210.00, 210.01–350.00, 350.01–700.00, >700.00 g/week) according to its quartiles. Similarly, aquatic/sea food product intake was also transformed into four categories (≤15.17, 15.18–93.33, 93.34–200.00, >200.00 g/week) according to its quartiles. Physical activity level (expressed as metabolic equivalent) was estimated based on self-reported activities using a Compendium of Physical Activities,[22] and the physical activities questionnaire referenced the global physical activity questionnaire. Nocturnal sleep and midday nap duration were collected according to the two questions: 'How many hours do you usually nocturnal sleep every day in the past month?' and 'How many minutes do you usually midday nap every day in the past month?'

Finally, the information regarding clinic variables was obtained including BMI, kidney function, dyslipidaemia,

diabetes and hypertension. BMI was calculated as weight divided by height$^2$ (kg/m$^2$) and grouped into '<24' '24–27.9' and '≥28'.[16] Kidney function was stratified five stages by estimated glomerular filtration rate (eGFR): chronic kidney disease 1 (CKD 1) (eGFR ≥90 mL/min), CKD 2 (eGFR 60–89 mL/min), CKD 3 (eGFR 30–59 mL/min), CKD 4 (eGFR 15–29 mL/min), CKD 5 (eGFR <15 mL/min) and eGFR was estimated by serum creatinine with the abbreviated Modification of Diet in Renal Disease equation formula (175× (serum creatinine (μmol/L)/88.4)$^{-1.234}$ × (age)$^{-0.179}$ × (0.79 if woman)).[23] Dyslipidaemia was defined using serum total cholesterol concentration of ≥6.2 mmol/L, triacylglycerol concentration of ≥2.3 mmol/L, low-density lipoprotein cholesterol concentration of ≥4.1 mmol/L, high-density lipoprotein cholesterol concentration of <1.0 mmol/L or a self-reported diagnosis of dyslipidaemia.[24] Diabetes was diagnosed as fasting blood glucose level of ≥7.0 mmol/L, glycosylated haemoglobin percentage of ≥6.5% or a self-reported diagnosis of diabetes.[25] Participants with were regarded as having hypertension if any of the following three items were met: systolic blood pressure of ≥140 mm Hg, diastolic blood pressure of ≥90 mm Hg or self-reported diagnosis of hypertension.[26]

### Statistical analysis

The analyses were performed with the SPSS, V.25.0. Characteristics of the participants, including sociodemographic characteristics, health-related behaviours and clinic variables, were compared according to the frequency of snoring ('no', 'occasionally' and 'frequently'). All continuous data were described as mean±SD, while categorical data were summarised as percentages. The one-way analysis of variance was used to compare the difference in continuous data, and $\chi^2$ test was used to compare the difference in categorical data. Then, the prevalence of hyperuricaemia with different snoring frequency was examined using the $\chi^2$ test in different age and gender groups. Finally, multivariable logistic regression analysis was performed to evaluate the relationship between snoring and hyperuricaemia after adjusting for different covariates in different age groups, gender groups, BMI groups and total population and OR and 95% CI for the outcome variable were calculated. Two-sided p value of <0.05 was considered to be statistically significant.

### Patient and public involvement

Patients and the public were not involved in the design of the study, recruitment and conduct of the study or dissemination of the study results.

## RESULTS
### Sample characteristics of participants

The characteristics of participants based on the categories of self-reported snoring are shown in table 1. The study composed of 22 389 individuals in the final analysis. Of the study subjects, 51.23%, 26.71%, 22.06% reported 'no snoring', 'snoring occasionally', 'snoring frequently',

respectively. Participants who snore were more likely to be aged 45–59, male, employed person, non-smokers, occasionally alcohol drinkers, non-tea drinkers. In addition, compared with those who reported 'no snoring', snorers had a higher level of energy take, midday nap duration, BMI, dyslipidaemia, diabetes, hypertension but had a lower level of physical activity, nocturnal sleep duration.

### The prevalence of hyperuricaemia

The prevalence of hyperuricaemia according to different snoring frequency with different groups is presented in table 2. In all, the prevalence of hyperuricaemia was 14.43% (3230/22389) in our study population. Compared with the 'no snoring' group, the prevalence of hyperuricaemia was higher in the 'snoring occasionally' and 'snoring frequently' groups among the three age groups (p<0.05). In addition, those who reported 'snoring occasionally' or 'snoring frequently' were more likely to have hyperuricaemia than 'no snoring' group in different gender groups (p<0.05). Similarly, in the total population, higher prevalence of hyperuricaemia was found in the 'snoring occasionally' or 'snoring frequently' compared with 'no snoring' group.

### Association between self-reported snoring and hyperuricaemia

Results of multivariable logistic regression analysis about the association between self-reported snoring and hyperuricaemia are shown in table 3. In the total population, after adjusting for sociodemographic characteristics, health-related behaviours and clinic variables, those who reported 'snoring occasionally' or 'snoring frequently' were more likely to be hyperuricaemia (OR 1.19, 95% CI 1.07 to 1.31; OR 1.33, 95% CI 1.19 to 1.47, respectively) compared with 'no snoring' group. The stratified analysis by age results showed: after adjusting for all variables, in three age groups higher OR of hyperuricaemia (OR 1.32, 95% CI 1.07 to 1.63; OR 1.23, 95% CI 1.04 to 1.45; OR 1.33, 95% CI 1.11 to 1.58, respectively) was found in the 'snoring frequently' group compared with 'no snoring' group. The stratified analysis by gender results showed: after adjusting for all variables, those who reported 'snoring frequently' had a higher OR of hyperuricaemia in men (OR 1.24, 95% CI 1.08 to 1.41) and woman (OR 1.33, 95% CI 1.12 to 1.59, respectively) than 'no snoring' group. The stratified analysis by BMI results showed: after adjusting for all variables, those who reported 'snoring frequently' remained at an increased risk of hyperuricaemia in BMI 24–27.9 and ≥28 kg/m$^2$ groups (OR 1.35, 95% CI 1.16 to 1.56; OR 1.41, 95% CI 1.15 to 1.73), but the association was not found in BMI <24 kg/m$^2$ group (p>0.05).

## DISCUSSION

In our study, we found that self-reported snoring was independently associated with hyperuricaemia in total population. After adjusting for the sociodemographic

**Table 1** Characteristics of participants based on the categories of self-reported snoring

| Characteristics | Self-reported snoring | | | P value |
| | No | Occasionally | Frequently | |
|---|---|---|---|---|
| Number of subjects | 11 469 | 5981 | 4939 | |
| **Sociodemographic characteristics** | | | | |
| Age | | | | <0.001 |
| <45 | 36.18 | 30.16 | 19.80 | |
| 45–59 | 39.64 | 42.37 | 43.59 | |
| >59 | 24.18 | 27.47 | 36.61 | |
| Male | 35.26 | 53.02 | 66.13 | <0.001 |
| Married or living with partner | 87.75 | 88.13 | 88.01 | 0.742 |
| Education level | | | | <0.001 |
| Primary or below | 33.16 | 28.57 | 36.83 | |
| Secondary | 49.94 | 53.10 | 49.20 | |
| College or above | 16.90 | 18.32 | 13.97 | |
| Urban | 27.49 | 33.42 | 25.81 | <0.001 |
| Employment | 63.05 | 62.43 | 59.73 | |
| Income (<60 000 RMB) | 59.51 | 55.12 | 59.91 | <0.001 |
| **Health-related behaviours** | | | | |
| Smoking | | | | <0.001 |
| Never | 80.99 | 69.27 | 59.71 | |
| Current | 14.61 | 23.61 | 31.00 | |
| Ever | 4.40 | 7.12 | 9.29 | |
| Alcohol drinking | | | | <0.001 |
| Never or hardly | 53.00 | 38.40 | 37.84 | |
| Occasionally | 37.56 | 45.23 | 40.27 | |
| Often | 9.44 | 16.37 | 21.89 | |
| Tea drinker | 16.27 | 24.33 | 28.10 | <0.001 |
| Sugar-sweetened beverage drinker | 1.96 | 2.51 | 3.52 | <0.001 |
| Oil intake (g/week) | 56.71±0.30 | 56.66±0.42 | 58.33+0.47 | 0.008 |
| Salt intake (g/week) | 48.36±0.27 | 47.45±0.36 | 51.04±0.42 | <0.001 |
| Energy intake (kcal/week) | 12776.57±39.74 | 13241.66±56.36 | 13838.48±64.34 | <0.001 |
| Red meat intake (g/week) | | | | <0.001 |
| ≤210.00 | 27.32 | 25.73 | 23.14 | |
| 210.01–350.00 | 26.06 | 25.10 | 21.83 | |
| 350.01–700.00 | 30.23 | 30.48 | 31.91 | |
| >700.00 | 16.39 | 18.69 | 23.12 | |
| Aquatic/sea food products intake (g/week) | | | | <0.001 |
| ≤15.17 | 27.10 | 22.07 | 23.87 | |
| 15.18–93.33 | 25.01 | 25.75 | 23.73 | |
| 93.34–200.00 | 25.38 | 26.25 | 24.13 | |
| >200.00 | 22.51 | 25.93 | 28.26 | |
| Physical activity (MET minutes/week) | 10176.77±63.98 | 9731.86±86.57 | 9777.99±101.00 | <0.001 |
| Nocturnal sleep duration (h/day) | 7.18±0.01 | 7.04±0.02 | 6.97±0.02 | <0.001 |
| Midday nap duration (min/day) | 28.91±0.33 | 31.17±0.45 | 33.43±0.54 | <0.001 |
| **Clinic variables** | | | | |
| BMI (kg/m$^2$) | | | | <0.001 |

Continued

**Table 1** Continued

| Characteristics | Self-reported snoring | | | P value |
| --- | --- | --- | --- | --- |
| | No | Occasionally | Frequently | |
| <24 | 55.07 | 39.26 | 24.20 | |
| 24–27.9 | 35.91 | 45.81 | 47.99 | |
| ≥28 | 9.02 | 14.93 | 27.82 | |
| Kidney function | | | | |
| CKD1 | 84.33 | 80.45 | 75.40 | <0.001 |
| CKD2 | 14.97 | 18.66 | 23.16 | |
| CKD3 | 0.66 | 0.80 | 1.28 | |
| CKD4 | 0.02 | 0.05 | 0.12 | |
| CKD5 | 0.02 | 0.03 | 0.04 | |
| Dyslipidaemia | 23.99 | 33.46 | 44.26 | <0.001 |
| Diabetes | 7.35 | 10.18 | 15.02 | <0.001 |
| Hypertension | 28.13 | 36.23 | 50.50 | <0.001 |

Continuous data were described as mean±SD, and statistical significance was assessed by the one-way analysis of variance.
Categorical data were summarised as percentages (%), and statistical significance was assessed by $\chi^2$ test.
BMI, body mass index; MET, metabolic equivalent.

characteristics, health-related behaviours and clinic variables, those who reported 'snoring occasionally' and 'snoring frequently' had 19% and 33% increased risk of hyperuricaemia compared with 'no snoring' group, respectively.

Our findings are consistent with those results of previous studies. An analysis of 6491 participants from the National Health and Nutrition Survey 2005–2008 found an increased risk of hyperuricaemia associated with snoring, which found similar results with our study.[15] Xiong *et al* enrolled 7699 Chinese urban adults from Nanjing between September 2016 and February 2018, they verified the positive link between snoring and hyperuricaemia.[16] A study of data from the baseline survey of the China Hainan Centenarian Cohort Study also showed a positive association between snoring and hyperuricaemia in centenarians.[12] The possible mechanisms can partly explain the positive association between snoring and hyperuricaemia. Snoring is often accompanied by oxygen desaturation, leading to tissue hypoxia, and then this accelerates the breakdown of ATP, resulting in the accumulation of UA from purine catabolic product by xanthine dehydrogenase and xanthine oxidase.[27] In

addition, previous study has shown that vibration induced by snoring is related to inflammation response,[28] otherwise some studies have found the association of inflammation response and metabolic disease, for example, dyslipidaemia.[29 30] Therefore, whether the effect of snoring on hyperuricaemia is mediated by inflammatory pathway needs to be further explored.

Compared with previous studies, our study conducted more detailed relationship analysis of snoring and hyperuricaemia in different age and gender groups. To our knowledge, no studies have been analysed age differences of association between snoring and hyperuricaemia. We found that those who snore frequently with aged >59 year old had a 33% increased risk of hyperuricaemia (p<0.05) compared with no snoring people, and the strength of association was stronger than age 45–59 (23%) and aged <45 (32%) groups, which might be related to decreasing metabolic function significantly in old people. This result suggested that people over 59 who snore frequently should pay more attention to their UA level in order to achieve early prevention of gout. In the present study, for men and women, those who snore frequently had a 24% and 33% increased risk of hyperuricaemia

**Table 2** Prevalence of hyperuricaemia according to snoring frequency in different groups

| Snoring frequency | Age (n (%)) | | | Gender (n (%)) | | Total |
| --- | --- | --- | --- | --- | --- | --- |
| | <45 | 45–59 | >59 | Male | Female | |
| No | 376 (9.06) | 441 (9.70) | 358 (12.91) | 650 (16.07) | 525 (7.07) | 1175 (10.25) |
| Occasionally | 326 (18.07) | 370 (14.60) | 261 (15.89) | 659 (20.78) | 298 (10.60) | 957 (16.00) |
| Frequently | 299 (30.57) | 452 (20.99) | 347 (19.19) | 820 (25.11) | 278 (16.62) | 1098 (22.23) |
| $\chi^2$ | 322.34 | 160.32 | 33.23 | 91.72 | 156.38 | 418.14 |
| P value | <0.001 | <0.001 | <0.001 | <0.001 | <0.001 | <0.001 |

**Table 3** Association between self-reported snoring and hyperuricaemia

| Variables | NO OR (95% CI) | Occasionally OR (95% CI) | P value | Frequently OR (95% CI) | P value |
|---|---|---|---|---|---|
| **Age** | | | | | |
| <45 | | | | | |
| Model 1 | 1.00 (reference) | 2.21 (1.89 to 2.60) | <0.001 | 4.42 (3.72 to 5.25) | <0.001 |
| Model 2 | 1.00 (reference) | 1.51 (1.28 to 1.79) | <0.001 | 2.44 (2.03 to 2.94) | <0.001 |
| Model 3 | 1.00 (reference) | 1.49 (1.26 to 1.77) | <0.001 | 2.38 (1.97 to 2.88) | <0.001 |
| Model 4 | 1.00 (reference) | 1.16 (0.96 to 1.39) | 0.13 | **1.32** (1.07 to 1.63) | **0.01** |
| 45–59 | | | | | |
| Model 1 | 1.00 (reference) | 1.59 (1.37 to 1.85) | <0.001 | 2.47 (2.14 to 2.85) | <0.001 |
| Model 2 | 1.00 (reference) | 1.33 (1.14 to 1.55) | <0.001 | 1.88 (1.62 to 2.18) | <0.001 |
| Model 3 | 1.00 (reference) | 1.29 (1.11 to 1.50) | 0.00 | 1.83 (1.57 to 2.12) | <0.001 |
| Model 4 | 1.00 (reference) | 1.09 (0.92 to 1.28) | 0.32 | **1.23** (1.04 to 1.45) | **0.02** |
| >59 | | | | | |
| Model 1 | 1.00 (reference) | 1.28 (1.07 to 1.52) | 0.01 | 1.60 (1.36 to 1.88) | <0.001 |
| Model 2 | 1.00 (reference) | 1.25 (1.05 to 1.48) | 0.01 | 1.59 (1.35 to 1.87) | <0.001 |
| Model 3 | 1.00 (reference) | 1.22 (1.03 to 1.46) | 0.03 | 1.58 (1.34 to 1.87) | <0.001 |
| Model 4 | 1.00 (reference) | 1.18 (0.98 to 1.42) | 0.09 | **1.33** (1.11 to 1.58) | **0.00** |
| **Gender** | | | | | |
| Male | | | | | |
| Model 1 | 1.00 (reference) | 1.37 (1.22 to 1.54) | <0.001 | 1.75 (1.56 to 1.96) | <0.001 |
| Model 2 | 1.00 (reference) | 1.30 (1.15 to 1.47) | <0.001 | 1.79 (1.59 to 2.00) | <0.001 |
| Model 3 | 1.00 (reference) | 1.25 (1.11 to 1.41) | <0.001 | 1.71 (1.52 to 1.93) | <0.001 |
| Model 4 | 1.00 (reference) | 1.08 (0.94 to 1.23) | 0.28 | **1.24** (1.08 to 1.41) | **0.00** |
| Female | | | | | |
| Model 1 | 1.00 (reference) | 1.56 (1.34 to 1.81) | <0.001 | 2.62 (2.24 to 3.06) | <0.001 |
| Model 2 | 1.00 (reference) | 1.42 (1.22 to 1.65) | <0.001 | 2.13 (1.81 to 2.50) | <0.001 |
| Model 3 | 1.00 (reference) | 1.43 (1.22 to 1.66) | <0.001 | 2.13 (1.81 to 2.51) | <0.001 |
| Model 4 | 1.00 (reference) | **1.21** (1.03 to 1.42) | **0.02** | **1.33** (1.12 to 1.59) | **0.00** |
| **BMI** | | | | | |
| <24 | | | | | |
| Model 1 | 1.00 (reference) | 1.48 (1.25 to 1.76) | <0.001 | 1.96 (1.61 to 2.40) | <0.001 |
| Model 2 | 1.00 (reference) | 1.22 (1.02 to 1.45) | 0.03 | 1.39 (1.12 to 1.71) | 0.00 |
| Model 3 | 1.00 (reference) | 1.20 (1.00 to 1.43) | 0.05 | 1.36 (1.10 to 1.68) | 0.00 |
| Model 4 | 1.00 (reference) | 1.13 (0.94 to 1.36) | 0.19 | 1.16 (0.93 to 1.45) | 0.19 |
| 24–27.9 | | | | | |
| Model 1 | 1.00 (reference) | 1.43 (1.25 to 1.64) | <0.001 | 1.84 (1.61 to 2.10) | <0.001 |
| Model 2 | 1.00 (reference) | 1.21 (1.05 to 1.39) | 0.01 | 1.50 (1.30 to 1.73) | <0.001 |
| Model 3 | 1.00 (reference) | 1.19 (1.03 to 1.37) | 0.02 | 1.48 (1.28 to 1.70) | <0.001 |
| Model 4 | 1.00 (reference) | 1.15 (0.99 to 1.33) | 0.06 | **1.35** (1.16 to 1.56) | **<0.001** |
| ≥28 | | | | | |
| Model 1 | 1.00 (reference) | 1.42 (1.16 to 1.74) | 0.00 | 1.68 (1.40 to 2.02) | <0.001 |
| Model 2 | 1.00 (reference) | 1.30 (1.06 to 1.61) | 0.01 | 1.47 (1.22 to 1.78) | <0.001 |
| Model 3 | 1.00 (reference) | 1.30 (1.05 to 1.61) | 0.02 | 1.47 (1.21 to 1.79) | <0.001 |
| Model 4 | 1.00 (reference) | 1.25 (1.00 to 1.55) | 0.05 | **1.41** (1.15 to 1.73) | **0.00** |
| Total | | | | | |

| | NO | Occasionally | | Frequently | |
|---|---|---|---|---|---|
| **Table 3** Continued | | | | | |
| **Variables** | **OR (95% CI)** | **OR (95% CI)** | **P value** | **OR (95% CI)** | **P value** |
| Model 1 | 1.00 (reference) | 1.67 (1.52 to 1.83) | <0.001 | 2.50 (2.29 to 2.74) | <0.001 |
| Model 2 | 1.00 (reference) | 1.45 (1.32 to 1.59) | <0.001 | 2.03 (1.84 to 2.23) | <0.001 |
| Model 3 | 1.00 (reference) | 1.43 (1.30 to 1.57) | <0.001 | 2.00 (1.81 to 2.20) | <0.001 |
| Model 4 | 1.00 (reference) | **1.19** (1.07 to 1.31) | **0.00** | **1.33** (1.19 to 1.47) | **<0.001** |

Model 1: unadjusted model.
Model 2: adjusted for gender (when stratification by gender, it was excluded), age (when stratification by age, it was excluded), marital status, education level, residential region, employment status and household yearly income level.
Model 3: additionally adjusted for smoking status, alcohol drinking status, tea drinking status, sugar-sweetened beverage drinking status, oil intake, salt intake, energy intake, red meat intake, aquatic/sea food products intake, physical activity level, nocturnal sleep and midday nap duration.
Model 4: additionally BMI, kidney function, dyslipidaemia, diabetes and hypertension.
Bold values indicate P value <0.05 in Model 4.
BMI, body mass index.

(p<0.05) than no snoring people, respectively. Our findings showed that the strength of association in women was stronger compared with men, which are consistent with previous study.[16] To the best of our knowledge, no study has explored the possible mechanisms. It might be related to hormone difference in women and men. Therefore, more evidence studies on the mechanism of different strength relationship between snoring and hyperuricaemia in women and mwn are needed. In this study, our results showed snoring occasionally and snoring frequently had increased risk of hyperuricaemia than no snoring people in all different age and gender groups before adjusting for clinic variables, although the association between snoring occasionally and no snoring was not found for all groups except for woman after adjusting for all confounders. This implies that BMI, kidney function, dyslipidaemia, diabetes and hypertension may affect the relationship between snoring frequency and hyperuricaemia in different age and gender groups. Previous studies had confirmed snoring to be significantly associated with obesity,[31] dyslipidaemia,[32] diabetes[33] and hypertension.[34] In addition, those diseases may be related to serum UA level.[35 36] Therefore, it is necessary to adjust clinical indicators.

Besides, we conducted the stratified analysis by BMI. The result revealed that those who snore frequently with a high BMI remained at an increased risk of hyperuricaemia compared with no snoring people, however, the positive association did not found in the low BMI group. The difference in results of three groups indicated the necessity to stratify by BMI. And this interesting result implies that overweight or obesity plays an important role in the relationship of snoring and hyperuricaemia, which may be related to hypoxemia caused by overweight or obesity.[37]

In addition, our results showed that the prevalence of hyperuricaemia was 14.43% in Chongqing, which was similar to the results of the Tianjin Brain Study (14.4%).[38] However, this prevalence was more than that in the Tangshan population (13.4%).[39] The difference may be explained by the different lifestyles and economic level in the two different areas, which are also closely related to hyperuricaemia.[40] Our determined prevalence was also slightly higher than that in southwest (Chengdu and Chongqing) China hyperuricaemia survey, which showed that the prevalence of hyperuricaemia was 13.5% in 2013–2014[10]. This indicates that we should pay more attention to hyperuricaemia due to the prevalence is increasing with the improvement of economic level and lifestyles changes.

Compared with previous study, our study is more thorough and comprehensive because the stratified analysis was conducted to explore the difference of relationship between snoring and hyperuricaemia in different age, gender and BMI category groups. Only one previous study conducted a stratified analysis by gender, but this study did not consider to stratified analysis by age and BMI category.[16] Moreover, participants of the prior study were recruited from hospital, so the sample populations were not representative. Relatively, community-based large samples of our study increased the representativeness and generalisability of our result to some extent. To the best of our knowledge, this is the first study to explore the association between snoring and hyperuricaemia used a large sample size in southwest China. Furthermore, the present study included many potential confounders, such as smoking, alcohol drinking, tea drinking, sugar-sweetened beverage drinking, oil intake, salt intake, energy intake, red meat intake, aquatic/sea food products intake, physical activity, sleep duration, nap duration, BMI, kidney function, dyslipidaemia, diabetes and hypertension, which could minimise the bias.

The present study also has some limitations. First, our study data were a cross-sectional result from the cohort study, and the cross-sectional data analysis limited the ability to explore the casual relationship between snoring and hyperuricaemia. Further follow-up data analysis could confirm whether there was causal relationship between

snoring and hyperuricaemia by using special statistical model. Second, snoring frequency was obtained from a subjective self-reported question, which may be different with actual snoring frequency. However, Jennum *et al* had found self-reported snoring frequency was highly correlated with microphone records,[41] suggesting that the self-reported snoring could be applied in scientific research. In addition, self-reported snoring manner has also been used to other similar studies of the relationship between snoring and UA.[12 15] Third, the present study was lack of degree of snoring, which may distort the relationship between snoring and hyperuricaemia. Thus, the degree of snoring should be considered in future studies. Fourth, our study was limited by age and ethnicity, so extrapolating the results to other age groups and ethnicity population should be noted.

## CONCLUSIONS

In summary, snoring frequency was positively associated with higher risk of hyperuricaemia in total population. In the three age groups, those who snore frequently had increased risk of hyperuricaemia compared with no snoring people, and the strength of association was the strongest in age >59 group. In addition, snoring frequently is an independent risk factor of hyperuricaemia for men and women, and the strength of association in women was stronger compared with men. In high BMI population, those who snore frequently had increased risk of hyperuricaemia compared with no snoring people, but the association was not found in BMI <24 group. Our result indicated that snoring frequently may be a signal for hyperuricaemia, especially for women, those over 59 years of age or those who are overweight or obese.

**Acknowledgements** We really appreciate all participants, project staff of the China Multi-Ethnic Cohort Study.

**Contributors** All authors contributed significantly to this article. TC analysed the data and wrote the first version of the manuscript; WT, LC, DM and XD performed the surveys; LS and XL interpreted the results and revised the manuscript; XD and XL critically reviewed the manuscript for important intellectual content; XL,XD and WT accepts full responsibility for the overall content as guarantor. All authors approved the final manuscript.

**Funding** This work was supported by National Key R&D Program of China (grant number: 2017YFC0907303) and Chongqing medical scientific research project (grant number: 2020FYYX013).

**Competing interests** None declared.

**Patient and public involvement** Patients and/or the public were not involved in the design, or conduct, or reporting, or dissemination plans of this research.

**Patient consent for publication** Not applicable.

**Ethics approval** This study involves human participants and was approved by the ethics committee of Sichuan University (Number K2016038). Participants gave informed consent to participate in the study before taking part.

**Provenance and peer review** Not commissioned; externally peer reviewed.

**Data availability statement** Data are available upon reasonable request.

**ORCID iD**
Ting Chen http://orcid.org/0000-0002-0723-6337

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
