## [Reviewer comments · BMJ Open]

ARTICLE DETAILS

TITLE (PROVISIONAL)	Association of self-reported snoring and hyperuricemia: a large cross-sectional study in Chongqing, China
AUTHORS	chen, ting; Ding, Xianbin; Tang, Wenge; Chen, Liling; Mao, Deqiang; Song, Lingling; Lian, Xuemei

VERSION 1 – REVIEW

REVIEWER	Natalie McCormick Massachusetts General Hospital, Division of Rheumatology, Allergy, and Immunology
REVIEW RETURNED	21-Nov-2021

GENERAL COMMENTS	Thank you for your efforts in preparing and submitting this manuscript on self-reported snoring and hyperuricaemia. Strengths of this study include the large, population-based sample of adults of a relevant age group, and array of clinical and lifestyle exposures included as covariates. Potential mechanisms are described in the Discussion; however, as acknowledged, the cross-sectional design limits the inferences that can be made. Please see some queries and suggestions below. 1. You mention in several places that prior studies in this area did not sufficiently control for potential confounders. Could you please elaborate, and describe the additional confounders that you included? It seems like the US NHANES study, for example, included many of the same covariates as you did, though perhaps not the dietary variables or other sleep characteristics. 2. Did you adjust for kidney disease or kidney function? A 2020 study using the UK CPRD found the overall association between sleep apnea and gout disappeared after adjustment for BMI, heart failure, and CKD. https://pubmed.ncbi.nlm.nih.gov/32334617/ 3. Given the strong association between BMI and sleep problems, as well as hyperuricaemia, could you please describe how BMI was adjusted, as was done for the other clinical variables – continuously? Categorical? If so, how many categories? 4. The sex-specific results were interesting, but additional subgroup analyses could make the study more innovative. For example, at least for sleep apnea and gout, there have been differences by BMI categories. 5. Abstract, final sentence and Conclusion, final sentence: Please consider editing as I don't feel the findings from this cross-sectional analysis provide enough justification about potentially intervening on snoring to prevent hyperuricaemia. Perhaps snoring could be used
---

	for screening? 6. Methods, page 6, paragraph 2, “The validity and accuracy of UA were 10% and 4%, respectively”: Could you please clarify what these statistics mean? 7. Discussion: Can you explain why the association was not present in younger-aged individuals? 8. Discussion, page 17, end of paragraph 1: Are data available on the longitudinal trends in hyperuricemia for the same population/data source? That would make your argument about increasing prevalence more compelling – the 14% reported for your region in 2018-2019 might not be comparable to the earlier figure you cited from the China National survey. 9. Discussion, page 17, paragraph 2: You could mention this point, about participants in one of the prior studies being recruited from a hospital, earlier-on as a feature of your study. 10. Discussion, top of page 18, “However, Jennum et al. had found the self-reported snoring frequency was highly significant related to snoring record by microphone objectively”: It would help to mention this in the Methods as well, aligning with your mentioning the validity of the serum urate assay. You could also discuss whether this (or a similar) questionnaire has been used in other studies – it looks like snoring frequency was ascertained in a similar manner in the NHANES paper you cited.
--	--

REVIEWER	Xiaoyan Zhang Zhongshan Hospital Fudan University, Department of nephrology
REVIEW RETURNED	23-Nov-2021

GENERAL COMMENTS	Dr. Ting Chen reported an association between self-reported snoring and hyperuricemia based on a large cross-sectional study in Chongqing, China. Dr. Chen performed questionnaire survey, physical examination and biological sample testing in 13 districts of Chongqing, including 22389 participants and proved that snoring frequency was positively associated with higher risk of hyperuricemia, and the strength of association varied with different age and gender. However, I would like that you address the following points: Major:  1. According to the updated meta-analysis “Demographic, regional and temporal trends of hyperuricemia epidemics in mainland China from 2000 to 2019: a systematic review and meta-analysis” [PMID: 33475474], the prevalence of hyperuricemia in southwestern China is as high as 21.2%, which may be attributed to regional eating habit, either is rich in purine content or interferences with the metabolism of purine. Did you take such eating habits into consideration in the questionnaire? 2. The authors should consider (if available) the fructose intake in the different group because fructose intake has been found to be strongly associated with SUA level [PMID: 31694231]. Minor:  1. The reference 1 needs to be updated. 2. Spelling mistakes and grammar errors. Page 5 Line 48 “13 district Chongqing”. Page 7 Line 12 “Sociodemographic characteristics
---

	included in our study age, gender ...". Page 9 Line 58 "middy". Page 21 Line 25 "vison". 3. The format of references need to be checked. Page 21 Line 27. 4. Page 8 51-53 "Participants who snore were older", but Table 1 showed the group aged 45-59 snore more, please check.
--	---

VERSION 1 – AUTHOR RESPONSE

Reviewer #1

1. Comment: You mention in several places that prior studies in this area did not sufficiently control for potential confounders. Could you please elaborate, and describe the additional confounders that you included? It seems like the US NHANES study, for example, included many of the same covariates as you did, though perhaps not the dietary variables or other sleep characteristics.

Response: Thank you for the suggestion. For the additional confounders, we included tea drinking, sugar-sweetened beverage drinking, oil intake, salt intake, energy intake, sleep duration, nap duration. The selection principle of these factors is mainly based on judgment of professional knowledge, literature research results and univariate analysis results in our study. These factors were associated with hyperuricemia and snoring status in univariate analysis. In addition, tea drinking, sugar-sweetened beverage drinking, oil intake, salt intake and energy intake may affect the metabolism of uric acid. Although sleep duration and nap duration have been found to be associated with serum uric acid level in the National Health and Nutrition Survey, the regression analysis did not include sleep and nap duration as covariates in the explore the relationship between snoring and high uric acid¹. Similarly, sugar-sweetened beverage drinking has been found to be associated with serum uric acid level in previous study², but no studies of snoring and hyperuricemia have considered sugar-sweetened beverage drinking. Accordingly, we have added those confounders in order to explore more accurate relationship of snoring and hyperuricemia.

On the other hand, to our knowledge, only three studies have been referred to explore relationship of snoring and hyperuricemia^{1 3 4}. In addition, the three studies had considered confounding factors insufficiently. The National Health and Nutrition Survey did not include some dietary variables and other sleep characteristics such as sleep duration, nap duration¹. The China Hainan Centenarian Cohort Study did not include alcohol drinking, physical activity, sleep duration, nap duration and other clinic variables such as dyslipidaemia, diabetes and hypertension³. The Chinese urban adults from Nanjing study did not include dietary variables, physical activity, sleep duration, and nap duration⁴. Compared with previous studies, almost all potential confounders were considered simultaneously in our study, which could minimize the bias and help us to explore more real relationship of snoring and hyperuricemia.

2. Comment: Did you adjust for kidney disease or kidney function? A 2020 study using the UK CPRD found the overall association between sleep apnea and gout disappeared after adjustment for BMI, heart failure, and CKD.

Response: Thank you for this valuable suggestion. We apologize that kidney function was missed in the manuscript, and we've adjusted the kidney function as covariate in the revised manuscript.

We reprocessed the data and presented them in tables 1 & 3 in revised vision. Generally, kidney function was estimated by glomerular filtration (eGFR). Furthermore, when only serum creatinine measurements were available, they were used to estimate the eGFR by the use of the abbreviated MDRD formula ⁵. Serum creatinine was measured in our study, and then eGFR was estimated by serum creatinine. In model 4 multivariable logistic regression analysis, we had adjusted all covariates including kidney function, which was stratified 5 stages by eGFR: CKD 1, CKD 2, CKD 3, CKD 4 and CKD 5. After adjusting for all variables including kidney function, snoring was positively associated with higher risk of hyperuricemia in the total population.

3. Comment: Given the strong association between BMI and sleep problems, as well as hyperuricemia, could you please describe how BMI was adjusted, as was done for the other clinical variables – continuously? categorical? If so, how many categories?

Response: Thank you for the suggestion. In our study, BMI was adjusted as categorical variable, which was grouped into “<24”, “24–27.9” and “≥28” kg/m². According to the criteria of overweight/obesity in China ⁶, overweight was defined as $24 \leq \text{BMI} < 28$ kg/m², and obesity was defined as $\text{BMI} \geq 28$ kg/m².

4. Comment: The sex-specific results were interesting, but additional subgroup analyses could make the study more innovative. For example, at least for sleep apnea and gout, there have been differences by BMI categories.

Response: Thank you very much for this reminder. We sincerely apologize for insufficient consideration, and we've added subgroup analysis by BMI in the revised manuscript.

We conducted the stratified analysis by BMI, which was grouped into three groups. The result revealed that those who snore frequently with a high BMI remained at an increased risk of hyperuricemia compared with no snoring people, however, the positive association did not found in the low BMI group. The difference in results of three groups indicated the necessity to stratify by BMI. And this interesting result implies that overweight or obesity plays an important role in the relationship of snoring and hyperuricemia, which may be related to hypoxemia caused by overweight or obesity ⁷.

The results have been presented in tables 3, and the discussion related to results had been added in the revised manuscript (discussion, paragraph 4). We hope that you find these revisions an improvement.

5. Comment: Abstract, final sentence and Conclusion, final sentence: Please consider editing as I don't feel the findings from this cross-sectional analysis provide enough justification about potentially intervening on snoring to prevent hyperuricemia. Perhaps snoring could be used for screening?

Response: Thank you very much for this suggestion. After our reconsideration, the previous statement about intervening on snoring to prevent hyperuricemia is indeed inappropriate, and we have revised the sentence using: "People who snore frequently should pay close attention to uric acid level in order to achieve early prevention of gout" in the revised manuscript. We hope that you find these revisions an improvement.

6. Comment: Methods, page 6, paragraph 2, "The validity and accuracy of UA were 10% and 4%, respectively": Could you please clarify what these statistics mean? UA.

Response: Thank you for this reminder. We apologize that the description about the validity and accuracy of serum UA assessment was unclear in the manuscript.

The validity of serum UA assessment means the degree of approximation between measured value and theoretical real value. Generally, the validity is described by relative deviation, if it is smaller, the validity is higher. The accuracy of serum UA assessment is described by coefficient of variations, which means the degree of variation of multiple measurements within the group.

The preceding descriptions have been added in the revised manuscript (Methods, Measures, section serum uric acid measurement) and we hope that our revisions are now more clearly explain the validity and accuracy of serum UA assessment.

7. Comment: Discussion: Can you explain why the association was not present in younger-aged individuals?

Response: Thank you for the suggestion. According to the suggestions of reviewers, we have added four covariates: kidney function, red meat intake, aquatic/sea food products intake, and sugar-sweetened beverage drinking in multivariable logistic regression analysis. After readjusting the variables, those who snore frequently had increased risk of hyperuricemia than no snoring people in all of the three age groups. We reprocessed the data and presented them in tables 1 & 3 in revised version.

8. Comment: Discussion, page 17, end of paragraph 1: Are data available on the longitudinal trends in hyperuricemia for the same population/data source? That would make your argument about increasing prevalence more compelling – the 14% reported for your region in 2018-2019 might not be comparable to the earlier figure you cited from the China National survey.

Response: Thank you very much for this suggestion. We sincerely apologize that there is no available data on the longitudinal trends in hyperuricemia for the same population/data source. The present study was the baseline survey of the China Multi-Ethnic Cohort Study, which had designed to duplicate survey, but it has not yet been conducted.

For exploring longitudinal trends, our determined prevalence was compared with that of the same region China hyperuricemia survey, which was conducted in southwest (Chengdu and Chongqing) China in 2013-2014⁸. Because Chongqing and Chengdu are adjacent cities, located in the southwest

of China, with similar living habit, environmental condition and economic level, the results of the two studies are comparable. Accordingly, we revised this section (discussion, paragraph 5) in revised version. We hope you find our revisions acceptable.

9. Comment: Discussion, page 17, paragraph 2: You could mention this point, about participants in one of the prior studies being recruited from a hospital, earlier-on as a feature of your study.

Response: Thank you very much for this suggestion. We have been mentioned this point, participants of the prior study were recruited from hospital with unrepresentative, and putted forward that community-based large samples of our study increased the representativeness and generalizability of our result to some extent in the revised manuscript. We hope that you find these revisions an improvement.

10. Comment: Discussion, top of page 18, “However, Jennum et al. had found the self-reported snoring frequency was highly significant related to snoring record by microphone objectively”: It would help to mention this in the Methods as well, aligning with your mentioning the validity of the serum urate assay. You could also discuss whether this (or a similar) questionnaire has been used in other studies – it looks like snoring frequency was ascertained in a similar manner in the NHANES paper you cited.

Response: Thank you very much for this suggestion. We have been added discussion content that self-reported snoring manner has also been used to other similar studies of the relationship between snoring and uric acid in the revised manuscript¹⁻³. We hope that you find these revisions an improvement.

Reviewer #2

Major revision:

1 Comment: According to the updated meta-analysis “Demographic, regional and temporal trends of hyperuricemia epidemics in mainland China from 2000 to 2019: a systematic review and meta-analysis”, the prevalence of hyperuricemia in southwestern China is as high as 21.2%, which may be attributed to regional eating habit, either is rich in purine content or interferences with the metabolism of purine. Did you take such eating habits into consideration in the questionnaire?

Response: Thank you very much for this suggestion. Although some dietary factors were taken into account such as alcohol drinking, tea drinking, oil intake, salt intake, energy intake, food categories rich in purine was missed in the manuscript. We sincerely apologize for insufficient consideration, and we’ve added red meat and aquatic/sea food products in the revised manuscript.

In this study, dietary habits were assessed using a semi-quantitative standardized food frequency questionnaire with 13 food categories, including rice, cooked wheaten food, coarse cereals, red meat, poultry, aquatic/sea food products, egg, fresh vegetables, bean products, preserved vegetables, fresh fruit, dairy product. Compared with other food, red meat and aquatic/sea food products are rich in purine content. Given the purine-rich food related to hyperuricemia, we performed regression analysis

adding red meat and aquatic/sea food products as covariates. Red meat intake was created as a dichotomized variable with 350 g/week as the cut-point according to median. Similarly aquatic/sea food products intake was also grouped into “≤93.33 g/week” and “>93.33 g/week”. Accordingly, we reprocessed the data and presented them in tables 1 & 3 in revised vision. We hope that you find these revisions an improvement.

2. Comment: The authors should consider (if available) the fructose intake in the different group because fructose intake has been found to be strongly associated with SUA level.

Response: Thank you for this valuable suggestion. We apologize that fructose intake was missed in the manuscript, and we've adjusted sugar-sweetened beverage, which is a main source of fructose, as covariate in the revised manuscript.

In the questionnaire, sugar-sweetened beverage drinking status had investigated by the question “Have you ever been drinking sugar-sweetened beverage weekly for more than half a year?” The participants were divided into drinker and non-drinker of sugar-sweetened beverage. Given sugar-sweetened beverage drinking possible related to hyperuricemia, we performed regression analysis adding sugar-sweetened beverage drinking status as covariate. Accordingly, we reprocessed the data and presented them in tables 1 & 3 in revised vision. We hope that you find these revisions an improvement.

Minor revision:

1. Comment: The reference 1 needs to be updated.

Response: Thank you for this reminding. We sincerely apologize for improper use of the reference in the manuscript. The reference 1 has been updated using new reference in revised manuscript⁹.

2. Comment: Spelling mistakes and grammar errors. Page 5 Line 48 “13 district Chongqing”. Page 7 Line 12 ” Sociodemographic characteristics included in our study age, gender ...”. Page 9 Line 58 “middy”. Page 21 Line 25 “vison”.

Response: Thank you for this reminding. We sincerely apologize for the spelling mistakes and grammar errors in the manuscript. The errors have been corrected in revised manuscript. We have carefully rechecked the manuscript to avoid any other similar mistakes.

3. Comment: The format of references need to be checked. Page 21 Line 27.

Response: Thank you for this reminding. We sincerely apologize for the format mistakes in the manuscript. The errors have been corrected in revised manuscript. We have carefully rechecked the manuscript to avoid any other similar mistakes.

4. Comment: Page 8 51-53”Participants who snore were older”, but Table 1 showed the group aged 45-59 snore more, please check.

Response: Thank you for this reminding. We sincerely apologize for the description about errors in the manuscript. The errors have been corrected in revised manuscript. We have carefully rechecked the manuscript to avoid any other similar mistakes.

VERSION 2 – REVIEW

REVIEWER	Natalie McCormick Massachusetts General Hospital, Division of Rheumatology, Allergy, and Immunology
REVIEW RETURNED	15-Jan-2022

GENERAL COMMENTS	Thank you very much for providing clarification, undertaking additional analysis, and revising this manuscript on the association between self-reported snoring and prevalent hyperuricemia. Strengths of this study include the large, population-based sample of adults of a relevant age group, and the many clinical and lifestyle exposures that were included in the models. You described potential mechanisms but the cross-sectional design limits the inferences that can be made about the development of hyperuricemia or gout. As such, I feel the Discussion and final sentence of the Abstract should be further revised; I suggest: “Frequent snoring may be a signal for hyperuricemia, especially for females, those over 59 years of age, or those who are overweight or obese. The additional stratification by BMI category and adjustment for kidney function and some dietary components are useful; however, I suggest using quintiles or multiple categories of consumption to more rigorously control for consumption of sugar-sweetened beverages and meats/seafood.
--

REVIEWER	Xiaoyan Zhang Zhongshan Hospital Fudan University, Department of nephrology
REVIEW RETURNED	11-Jan-2022

GENERAL COMMENTS	 1. the MDRD-CN formula ($175 \times \text{Scr}^{-1.234} \times \text{Age}^{-0.179} [\times 0.79 \text{ if female}]$, where Scr is serum creatine) might be more suitable for eGFR estimation in Chinese population. 2. Grammar errors. Page 13 Line 28 and 33 Association self-reported snoring and hyperuricemia should be added a prep. “between”. Page 17 Line 19-20, “In addition, some studies have showed that snoring is related to the inflammation response by vibration tissue cell releasing inflammatory factor”. Page 18 Line 33-35 “studies had confirmed snoring to be significantly association with” Should be corrected to “be significantly associated with”. Page 20 Line 19-20 “limited ability” should be corrected to “limited the ability”. 3. Page 16 Line 17 and 18 Model 2: adjusted for gender (when stratification by gender, it was included), age (when stratification by age, it was included). Included or excluded? 4. Reference 28 is the mechanism of snoring in children, who are frequently diagnosed in association with adenotonsillar hypertrophy. But adenotonsillar hypertrophy is mostly diagnosed among children. We suggest to remove this reference.
---

VERSION 2 – AUTHOR RESPONSE

Reviewer #1

1. Comment: I feel the Discussion and final sentence of the Abstract should be further revised; I suggest: “Frequent snoring may be a signal for hyperuricemia, especially for females, those over 59 years of age, or those who are overweight or obese”.

Response: Thank you for this valuable suggestion. We have further revised previous statement using: “Snoring frequently may be a signal for hyperuricemia, especially for females, those over 59 years of age, or those who are overweight or obese” in the Discussion and Abstract of revised manuscript.

2. Comment: The additional stratification by BMI category and adjustment for kidney function and some dietary components are useful; however, I suggest using quintiles or multiple categories of consumption to more rigorously control for consumption of sugar-sweetened beverages and meats/seafood.

Response: We appreciate your sincere suggestions. Red meat intake and aquatic/sea food products intake were retransformed into four categories according to its quartiles, and we reprocessed the data and presented them in tables 1 & 3 in revised vision. For sugar-sweetened beverages, the survey results showed that only 2.5% people (549/22389) had the habit of drinking sugar-sweetened beverage, which was a very low proportion. If further multiple categories of sugar-sweetened beverages were conducted, we are afraid that this may lead to some statistical bias. Therefore, sugar-sweetened beverages drinking status was dichotomous variable according to whether they had the habit of drinking.

Reviewer #2

1. Comment: the MDRD-CN formula ($175 \times \text{Scr}^{-1.234} \times \text{Age}^{-0.179} [\times 0.79 \text{ if female}]$, where Scr is serum creatine) might be more suitable for eGFR estimation in Chinese population.

Response: Thank you for your kindly suggestion. We have recalculated glomerular filtration (eGFR) using the MDRD-CN formula ($175 \times (\text{serum creatinine } (\mu\text{mol/L})/88.4)^{-1.234} \times (\text{age})^{-0.179} \times (0.79 \text{ if female})$), and kidney function was stratified 5 stages by recalculated eGFR. Further we reprocessed the data and presented them in tables 1 & 3 in revised vision.

2. Comment: Grammar errors. Page 13 Line 28 and 33 Association self-reported snoring and hyperuricemia should be added a prep. “between”. Page 17 Line 19-20, “In addition, some studies have showed that snoring is related to the inflammation response by vibration tissue cell releasing inflammatory factor”. Page 18 Line 33-35 “studies had confirmed snoring to be significantly association with” Should be corrected to “be significantly associated with”. Page 20 Line 19-20 “limited ability” should be corrected to “limited the ability”.

Response: Thanks for your careful checks. We sincerely apologize for the grammar errors in the manuscript. The errors have been corrected in revised manuscript. We have carefully rechecked the manuscript to avoid any other similar mistakes.

3. Comment: Page 16 Line 17 and 18 Model 2: adjusted for gender (when stratification by gender, it was included), age (when stratification by age, it was included). Included or excluded?

Response: Thank you for this reminding. We sincerely apologize for the spelling mistakes in the manuscript. The error has been corrected in revised manuscript.

4. Comment: Reference 28 is the mechanism of snoring in children, who are frequently diagnosed in association with adenotonsillar hypertrophy. But adenotonsillar hypertrophy is mostly diagnosed among children. We suggest to remove this reference.

Response: Thank you very much for your sincere suggestion. We have removed this reference in revised manuscript.

Again, we appreciate all your insightful comments. Thank you for taking the time and energy to help us improve the paper.

VERSION 3 – REVIEW

REVIEWER	Xiaoyan Zhang Zhongshan Hospital Fudan University, Department of nephrology
REVIEW RETURNED	08-Mar-2022
GENERAL COMMENTS	The reviewer completed the checklist but made no further comments.